# Neural mediation of greed personality trait on economic risk-taking

**Weiwei Li[1], Haixia Wang[2], Xiaofei Xie[3], Jian Li[3]***

[1]Academy for Advanced Interdisciplinary Studies, Peking University, Beijing, China; [2]School of Management, Jinan University, Guangzhou, China; [3]School of Psychological and Cognitive Sciences and Beijing Key Laboratory of Behavior and Mental Health, Peking University, Beijing, China

**Abstract** Dispositional greed, characterized by the insatiable hunger for more and the dissatisfaction for not having enough, has often been associated with heightened impulsivity and excessive risk-taking. Despite its far-reaching implications in social sciences and economics, however, the exact neural mechanisms of how greed personality influences risk-taking are still ill understood. In the present study, we showed the correlation between subject's greed personality trait (GPT) score and risk-taking was selectively mediated by individual's loss aversion, but not risk attitude. In addition, our neuroimaging results indicated that gain and loss prospects were jointly represented in the activities of the ventral striatum and medial orbitofrontal cortex (mOFC). Furthermore, mOFC responses also encoded the neural loss aversion signal and mediated the association between individual differences in GPT scores and behavioral loss aversion. Our findings provide a basis for understanding the specific neural mechanisms that mediate the effect of greed personality trait on risk-taking behavior.

DOI: https://doi.org/10.7554/eLife.45093.001

## Introduction

Ancient Greek philosopher Socrates allegedly said of greed that 'He who is not contented with what he has, would not be contented with what he would like to have'. Interestingly, recent development of greed research also emphasizes two parallel facets in the operational definition of greed personality trait: the desire of always wanting more and the dissatisfaction of not having enough (*Krekels and Pandelaere, 2015*; *Mussel and Hewig, 2016*; *Mussel et al., 2015*; *Seuntjens et al., 2015*). Greed has long been considered a double-edged sword: it motives people to compete and outperform for better well-being, but is also associated with negative stereotypes such as immorality (*Livingstone and Lunt, 1992*; *Lunt and Livingstone, 1991*; *Piff et al., 2012*; *Rokeach, 1973*; *Seuntjens et al., 2016*; *Tickle, 2004*; *Vohs et al., 2006*; *Zandi, 2008*) and excessive risk-taking which has been accused for insurmountable debt level and recent meltdown in the finance industry (*Aquino et al., 2009*; *Lo et al., 2005*). Understanding how greed might influence human risk-taking behavior, therefore, has far-reaching implications in cognitive and social sciences.

Although anecdotes and field studies generally suggest a potential link between greed personality trait and individual risk-taking, current research demonstrating this association is much less comprehensive and the results are inconsistent (*Mussel et al., 2015*; *Seuntjens et al., 2015*). For example, it has been shown that the self-assessed greed personality score was not related to risk-taking in a gain-only environment (*Seuntjens et al., 2015*), while other studies tended to suggest that greed personality trait significantly influences individual's risky taking behavior. In addition, recent Electroencephalography (EEG) studies also showed reduced feedback-related negativity-difference to losses relative to gains in human subjects high in trait-greed (*Mussel and Hewig, 2016*; *Mussel et al., 2015*). In light of these findings, one potential hypothesis to reconcile the conflicts in

*For correspondence:
leekin@gmail.com

Competing interests: The authors declare that no competing interests exist.

the extant literature is that greed personality trait might influence risk-taking along specific dimension that is not ubiquitous to all the risky decision-making tasks. For instance, subjects' risk-taking is believed to be driven by their risk attitudes in the gain or loss only gambling tasks, but is influenced by both risk attitude and relative sensitivity towards gains versus losses (loss aversion sensitivity) in a gain-loss mixed gambling task. This hypothesis can be formalized under the theoretical framework of prospect theory, where risk attitude ($\alpha$) and loss aversion coefficient ($\lambda$) together determine the degree of risk-taking in a value-based gambling task (*Barkley-Levenson et al., 2013*; *Kahneman and Tversky, 1979*; *Köbberling and Wakker, 2005*; *Tversky and Kahneman, 1992*). Therefore, extant literatures raise an intriguing possibility that individual greed personality trait might be related to the degree of loss aversion, but not risk attitude. It further predicts that greed personality would only affect risk-taking in mixed gamble tasks, where loss aversion plays a role (*Mussel et al., 2015*), but not in gain/loss only tasks, where loss aversion is not involved or canceled out (*Seuntjens et al., 2016*; *Seuntjens et al., 2015*). Here using functional magnetic resonance imaging (fMRI) with a classic gambling task, we directly tested this hypothesis and investigated whether and how greed personality trait might influence subjects' risky choice behavior. Previous research suggested that the activities of a network of brain structures, including ventral striatum, amygdala, and OFC, might encode a 'neural loss aversion' signal (*Canessa et al., 2013*; *De Martino et al., 2010*; *De Martino et al., 2009*; *Park et al., 2011*; *Plassmann et al., 2010*; *Tom et al., 2007*). We thus focused on these brain structures and examined how their responses were related to subjects' trait-greed, risk attitude and loss aversion. Finally, given that one significant concomitant personality trait often associated with greed is impulsivity (*Seuntjens et al., 2015*) and impulsivity has been implicated to influence risk-taking (*Baumann and Odum, 2012*; *Lejuez et al., 2003*; *Pack et al., 2001*; *Zaleskiewicz, 2001*), we also measured subject's impulsivity trait score to dissociate it from the effect of trait-greed in risk-taking behavior.

## Results

We first ran a pilot online study asking participants to fulfill the Dispositional Greed Scale (DGS) (*Seuntjens et al., 2015*) and Barratt Impulsiveness Scale (BIS-11) (*Patton et al., 1995*) to measure greed personality trait (GPT) and impulsivity personality trait (IPT) and tested the potential relationship between their GPT and IPT scores, as previous study suggested (*Seuntjens et al., 2015*). In the fMRI experiment, in addition to the completion of GPT and IPT measurements (see Materials and methods for details), in order to examine whether the greed personality trait scores significantly influence individuals' risky behavior, each subject completed a gambling task consisting of gain, loss and mixed trials while their brain activities were simultaneously monitored in the fMRI scanner. In each trial, subjects were asked to make a choice between options of a risky gamble and a guaranteed sure amount (see experiment procedure in *Figure 1* and gamble sets in *Figure 1—figure supplement 1*).

### GPT and task performance

Since impulsivity was also considered a potential factor influencing people's risky choices (*Clark et al., 2013*; *Studer and Clark, 2011*), especially among adolescents, pathological gamblers and drug addicts (*Barkley-Levenson and Galván, 2014*; *Chandler et al., 2009*; *Pehlivanova et al., 2018*; *Verdejo-García et al., 2008*), we ran the simple regression analysis between GPT and IPT scores both in the pilot and fMRI datasets. Consistent with previous studies (*Seuntjens et al., 2015*), we found subjects' GPT and IPT scores significantly correlated with each other in the pilot questionnaires ($r = 0.272$, $p = 0.018$) as well as in the fMRI dataset ($r = 0.543$, $p = 0.005$; *Figure 2A*), suggesting that IPT might be a confounding factor in identifying the specific effect of trait-greed (*Supplementary file 2*). Therefore, unless mentioned otherwise, we controlled the effect of subject's impulsivity by including the IPT score as a covariate of no interest in the following behavioral and imaging analyses.

We tested whether subjects' GPT scores were related to their gambling propensities among three different gamble types (gain, loss, and mixed trials). Our data showed that the GPT was significantly correlated with the percentage of risky choices in the mixed gambles ($r = 0.440$, $p = 0.032$; *Figure 2B*), but not in gain ($r = 0.236$, $p = 0.267$; *Figure 2C*) nor loss only gambles ($r = -0.002$, $p = 0.992$; *Figure 2D*). Furthermore, corroborating previous research that impulsivity was mainly

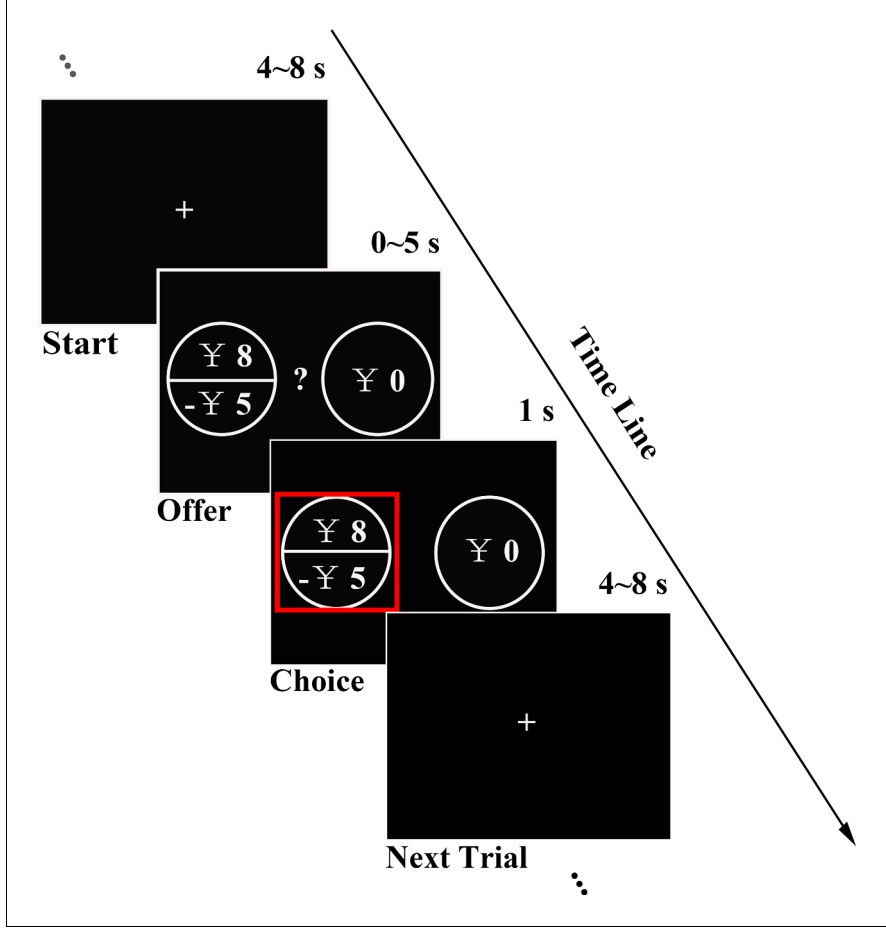

**Figure 1.** Experimental design. In each trial, subjects had 5 s to decide between a certain option and a fair gamble with gain, loss or mixed prospects. Chosen option was highlighted with a red rectangle for 1 s before a random ITI (4 ~ 8 s) was introduced.

DOI: https://doi.org/10.7554/eLife.45093.002

The following figure supplement is available for figure 1:

**Figure supplement 1.** Choice sets and average percentage of risky choices across subjects.

DOI: https://doi.org/10.7554/eLife.45093.003

related to the preference bias in the delay discounting tasks, our subjects' IPT scores were not correlated with the percentages of risky choices in all three-gamble contexts (see *Figure 2—figure supplement 1*). Importantly, we replicated these results by recruiting an independent cohort of subjects and asked them to perform the same behavioral task (*Figure 2—figure supplement 2*). The discrepancy of correlation results between GPT and gambling percentages among three tasks implied that the GPT might impact subjects' risk-taking only when both prospective gains and losses were presented (mixed trials) but not when single valence trials were involved (gain and loss only).

It is possible that above results were mainly driven by the fact that there were more trials in the mixed condition than the gain/loss only conditions and therefore led to a more robust estimation in the mixed condition. To exclude this possibility, we adopted a more rigorous model-based approach by specifying a prospect theory model to capture individual subject's behavior (*Kahneman and Tversky, 1979*; *Sokol-Hessner et al., 2013*; *Sokol-Hessner et al., 2009*; *Tversky and Kahneman, 1992*). Importantly, according to the prospect theory model, individual risk-taking was driven solely by the risk attitude parameter ($\alpha$, a constant representing valence sensitivity) in the gain and loss only trials, but by the interplay of both risk attitude ($\alpha$) and loss aversion ($\lambda$, a parameter capturing the asymmetrical weighting of loss relative to gain) parameters in the mixed condition (see Materials and methods for details). We took the hierarchical Bayesian analysis (HBA) approach in model fitting

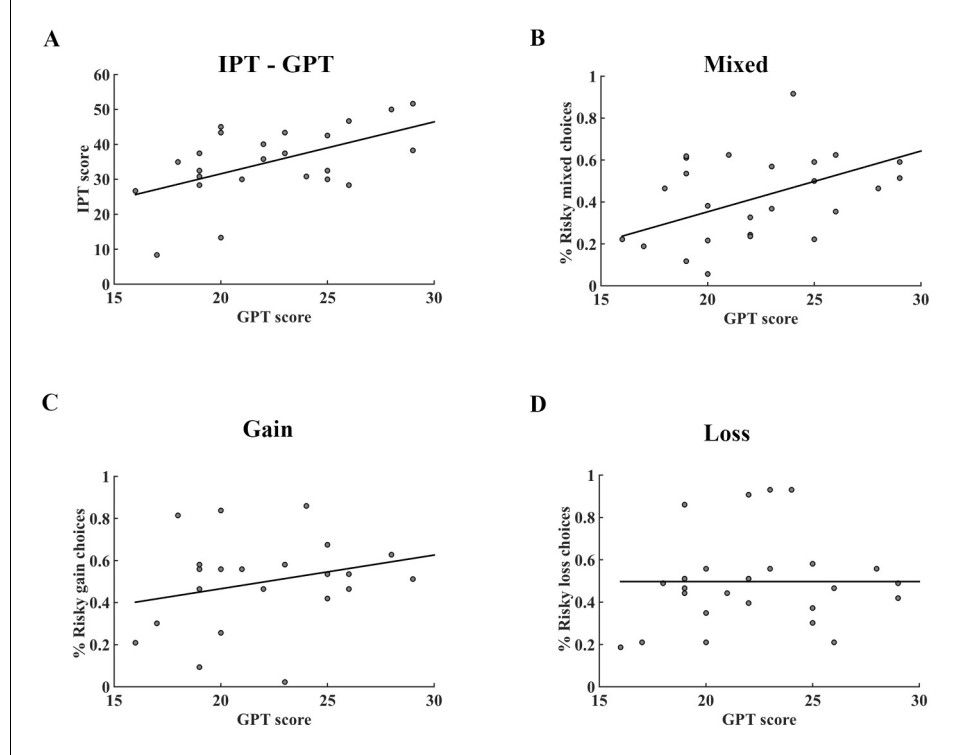

**Figure 2.** Correlation between GPT and IPT scores, GPT score and the percentage of risky choices in different conditions of the fMRI dataset. (**A**) There was a significant positive correlation between GPT and IPT scores ($r = 0.543$; $p = 0.005$). (**B**) – (**D**) The association between GPT score and the percentage of risky choices in mixed, gain, loss tasks, respectively. The GPT score was significantly correlated with the percentage of risky choices in (**B**) mixed trials ($r = 0.440$, $p = 0.032$), but not in (**C**) gain ($r = 0.236$, $p = 0.267$) nor (**D**) loss trials ($r = -0.002$, $p = 0.992$). GPT, greed personality trait; IPT, impulsivity personality trait.

DOI: https://doi.org/10.7554/eLife.45093.004

The following figure supplements are available for figure 2:

**Figure supplement 1.** The relationship between IPT score and the percentage of risky choices in the fMRI dataset.
DOI: https://doi.org/10.7554/eLife.45093.005

**Figure supplement 2.** In the behavioral replication task, the correlation between personality trait scores and the percentage of risky choices in different conditions.
DOI: https://doi.org/10.7554/eLife.45093.006

to obtain the estimated model parameters for individual subjects (*Ahn et al., 2013*). Consistent with previous literature suggesting that people weigh losses more than gains (*Chib et al., 2012*; *De Martino et al., 2010*; *Sokol-Hessner et al., 2009*; *Tom et al., 2007*), our results showed that the average loss aversion parameter ($\lambda$; mean = 1.561) across all subjects was significantly larger than 1 ($t = 3.874$, $p = 0.001$), implicating general loss aversion across subjects. Average risk attitude was not significantly different from 1 (mean = 1.033, $t = 0.812$, $p = 0.425$), indicating risk neutrality (see *Supplementary file 1*). Additionally, risk attitude and loss aversion parameters were not correlated ($r = 0.020$, $p = 0.925$).

We then examined whether the loss aversion and risk attitude parameters were associated with GPT scores across individuals using linear regression. Our results showed that subjects' loss aversion coefficients ($\lambda$) were negatively correlated with their GPT scores ($r = -0.479$, $p = 0.018$; *Figure 3A*), but no correlation was found between risk attitude coefficients ($\alpha$) and GPT scores ($r = 0.062$, $p = 0.775$; *Figure 3B*). Thus, our model-based analyses confirmed previous behavioral results (*Figure 2B–D*) and suggested that individual GPT score was specifically tied to loss aversion coefficient but not subject's risk attitude. In contrast, participants' IPT scores were not correlated with $\alpha$ nor $\lambda$ parameters of the model (*Figure 3—figure supplement 1*; *Supplementary file 1*). Furthermore, we found a significant negative correlation between loss aversion coefficient and the

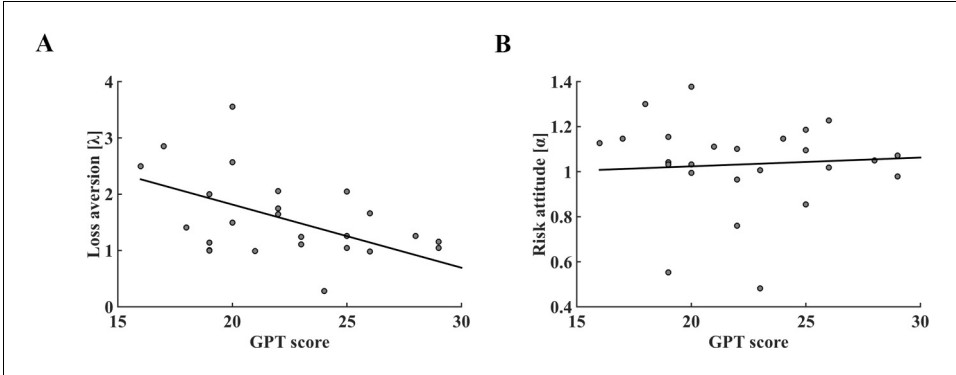

**Figure 3.** The relationship between GPT score and parameters of prospect theory decision-making model in the fMRI dataset. (A) Individual GPT score was significantly associated with model derived loss aversion coefficient ($r = -0.479$, $p = 0.018$). (B) Correlation between GPT score and risk attitude was not significant ($r = 0.062$, $p = 0.775$). GPT, greed personality trait.

DOI: https://doi.org/10.7554/eLife.45093.007

The following figure supplements are available for figure 3:

**Figure supplement 1.** The relationship between IPT score and the parameters of decision-making model based on prospect theory in the fMRI dataset.

DOI: https://doi.org/10.7554/eLife.45093.008

**Figure supplement 2.** Behavioral loss aversion mediated the effect of GPT on the percentage of risky choices (% risky choices) in the fMRI dataset.

DOI: https://doi.org/10.7554/eLife.45093.009

**Figure supplement 3.** In the behavioral replication task, the relationship between personality trait scores and parameters of prospect theory decision-making model.

DOI: https://doi.org/10.7554/eLife.45093.010

**Figure supplement 4.** In the behavioral replication task, the loss aversion parameter fully mediated the relationship between GPT and the percentage of risky choices (% risky choices).

DOI: https://doi.org/10.7554/eLife.45093.011

---

percentage of risky gamble choice across all trials ($r = -0.919$, $p < 0.001$), and individual subject's loss aversion coefficient ($\lambda$) fully mediated the correlation between GPT score and risky choice percentage across subjects (indirect effect: a × b = 0.024, 95% CI [0.007 0.045]; direct effect: c'=−0.003, 95% CI [−0.014 0.008]) (*Figure 3—figure supplement 2*) (*Preacher and Hayes, 2008*; *Zhao et al., 2010*). Together, our behavioral results showed that subjects' greed personality traits were associated with their risky choice preference and suggested loss aversion as the potential mediator for such effect (Similar results were found in the behavioral replication study, see *Figure 3—figure supplements 3–4*).

## Brain–behavior relationship

For imagining analyses, we first examined how prospective gains and losses were represented in the brain activities. We constructed a general linear model (GLM) at the onset of option revelation and entered the magnitudes of potential gain and loss in the gamble option and the outcome of sure option as parametric modulators for each subject. By design, the correlation between the magnitudes of potential gain and loss in the gamble option was not significant across trials ($r = 0.011$, $p = 0.867$), allowing us to investigate their corresponding neural activities separately. Consistent with previous studies (*Canessa et al., 2013*; *Park et al., 2011*; *Plassmann et al., 2010*; *Tom et al., 2007*), we found a neural network including bilateral orbitofrontal cortex (OFC), bilateral parietal cortex, and left posterior cingulate, whose activities positively correlated with potential gains of the gambles ($z = 3.1$, $p < 0.05$ false discovery rate (FDR) cluster correction; *Figure 4A*). Similarly, brain activities in the right OFC, bilateral striatum, bilateral insula and bilateral inferior parietal gyrus were found negatively correlated with the magnitudes of the potential loss associated with the gamble ($z = 3.1$, $p < 0.05$ FDR correction; *Figure 4B*; *Supplementary file 3*). Furthermore, our conjunction

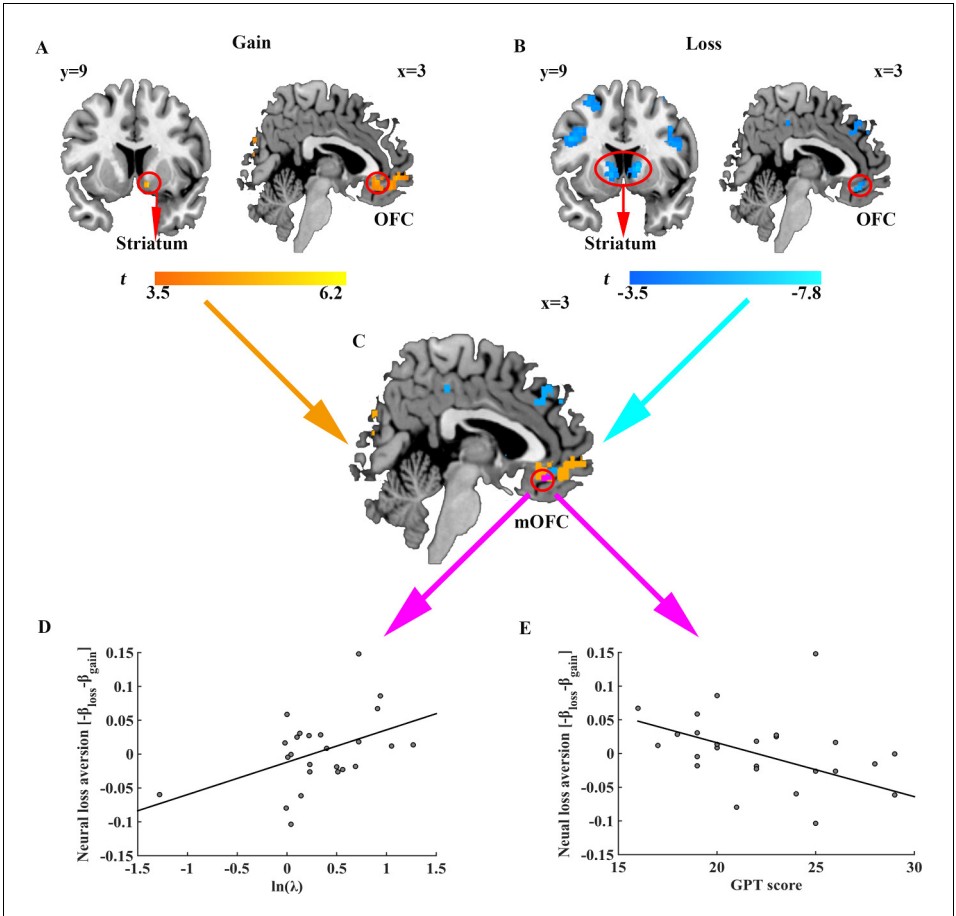

**Figure 4.** Neural representation of gain and loss prospects, behavioral loss aversion and GPT score. Brain regions whose activities (**A**) positively correlated with gain prospects and (**B**) negatively correlated with loss prospects. Results are shown uncorrected (p<0.001, voxel size >20) for visualization purposes. (**C**) Brain areas encoding both gain and loss prospects (purple). (**D**) Scatter plot indicating significant correlation between behavioral loss aversion (ln (λ)) and neural loss aversion (-$\beta_{loss}$ - $\beta_{gain}$) in the mOFC (r = 0.450, p = 0.024). (**E**) Scatter plot showing the significant correlation between GPT score and neural loss aversion (-$\beta_{loss}$ - $\beta_{gain}$) in the mOFC (r = −0.449, p = 0.028).

DOI: https://doi.org/10.7554/eLife.45093.012

The following figure supplement is available for figure 4:

**Figure supplement 1.** Ventral striatum (circled) activities also encoded gain and loss prospects, but with a more lenient threshold (z = 2.3, p < 0.05 FDR corrected).

DOI: https://doi.org/10.7554/eLife.45093.013

analysis identified that the medial OFC (mOFC) activity was related to the representation of both gain and loss prospects of the gamble (*Figure 4C*). Consistent with previous literature (*Tom et al., 2007*), this conjunction analysis also revealed an overlapping area in the ventral striatum, albeit at a more lenient threshold (z = 2.3, p < 0.05 FDR correction; *Figure 4—figure supplement 1*).

Previous literatures focusing on brain-behavior correlation suggested that the neural structures related to loss aversion should be sensitive to both gain and loss prospects and indicated limbic structures such as ventral striatum and OFC to encode individual behavioral loss aversion (*Canessa et al., 2013*; *Tom et al., 2007*). To test this hypothesis, we performed a region of interest (ROI) analysis using the mOFC mask obtained from the aforementioned conjunction analysis and tested whether BOLD responses in the mOFC correlated with behavioral loss aversion across subjects. Indeed, neural loss aversion in the mOFC, defined as the difference between mOFC activities towards losses versus gains (-$\beta_{loss}$ - $\beta_{gain}$) (*Tom et al., 2007*), showed a significant positive correlation with behavioral loss aversion coefficients (ln (λ); r = 0.450, p = 0.024) even after correction for

potential outliers (robust regression, $t = 2.203$, $p = 0.038$; *Figure 4D*). Moreover, as expected, such neural loss aversion signal also negatively correlated with individual subject's GPT score ($r = -0.449$, $p = 0.028$; robust regression, $t = -2.239$, $p = 0.035$), echoing our behavioral results of the association between GPT scores and behavioral loss aversion (ln ($\lambda$); $r = -0.434$, $p = 0.034$). Finally, activities in the ventral striatum, also identified from conjunction analysis above, did not exhibit significant correlation with behavioral loss aversion coefficients (ln ($\lambda$); $r = -0.012$, $p = 0.954$) nor with GPT score ($r = -0.239$, $p = 0.262$).

Given that the neural loss aversion measures in the mOFC correlated with both individuals' GPT scores and behavioral loss aversion coefficients (ln ($\lambda$)), we reasoned that personal greed personality trait (GPT score) might influence subject's behavioral loss aversion via the mediation of the mOFC activity. To test this hypothesis, we performed a mediation analysis where (ln ($\lambda$)) is the task related dependent variable, GPT scores the input variable, and the neural loss aversion in the mOFC the mediator. We controlled for potential confounds by including variables such as individuals' impulsivity scores (IPT) as covariates of no interest in the regression analysis (*Preacher and Hayes, 2008*; *Zhao et al., 2010*) (see Materials and methods for details). Consistent with our prediction, we found the correlation between individual's trait-greed and behavioral loss aversion could be fully mediated by the neural loss aversion measures in the mOFC (indirect effect: a × b = $-0.024$, 95% CI [$-0.084 - 0.001$]; direct effect: c'=$-0.047$, 95% CI [$-0.117$ 0.024]), suggesting higher GPT score (greed personality trait) might lead to lower behavioral loss aversion (task performance) by engaging brain activities in the mOFC, which by itself is a pivotal structure involved in the integration of gain and loss prospects (*Figure 5*).

## Discussion

Greed, hailed by some and dismissed by others, has long been recognized as a critical factor influencing people's risk-taking behavior (*Krekels and Pandelaere, 2015*; *Mussel et al., 2015*; *Seuntjens et al., 2015*). Traditional wisdom and recent developments in social psychology emphasize two components of greed: insatiable hunger for more and dissatisfaction for not having enough. In a standard risky decision making context, different greed components might be related to risk-

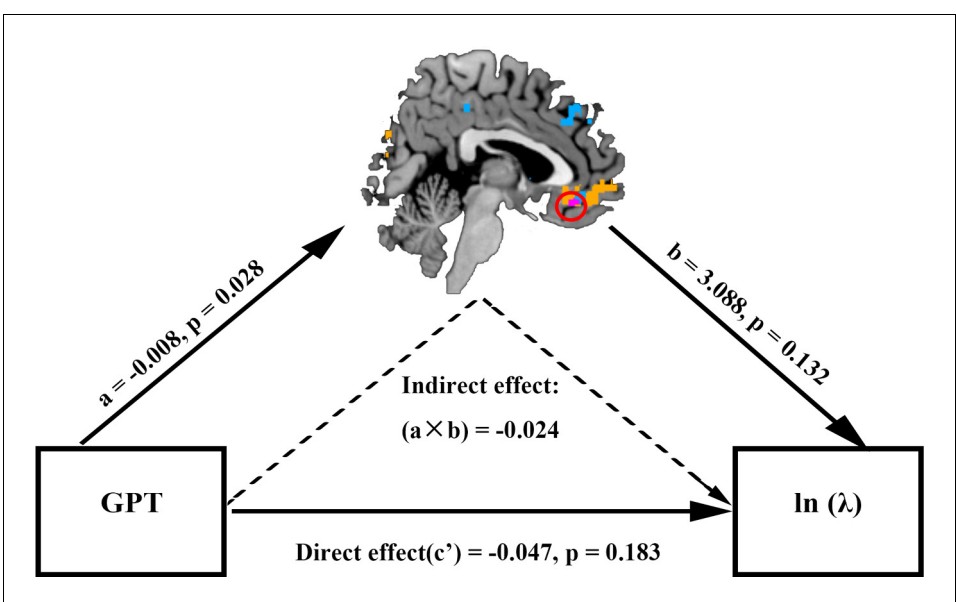

**Figure 5.** The mOFC neural loss aversion signal mediated the effect of GPT on behavioral loss aversion. Path a represented effect of GPT score on mOFC activity; path b indicated effect of mOFC signal on behavioral loss aversion; path c' represented the direct effect of greed personality on behavioral loss aversion when the mOFC activity was included as a mediator. The indirect effect (a × b) was significant with 95% bootstrap CI [$-0.084 - 0.001$] and the direct effect (c') did not reach significance (95% CI [$-0.117$ 0.024]), indicating full mediation.
DOI: https://doi.org/10.7554/eLife.45093.014

taking via influencing subject's risk attitude and/or loss aversion/gain seeking preference (*Sokol-Hessner et al., 2009*). Indeed, insatiable appetite for more is associated with elevated impulsivity and novelty seeking, which are comorbid with obsessive gambling, substance and behavioral addiction and aberrant behavior in adolescence (*Clark, 2014*; *Robbins and Clark, 2015*; *Seuntjens et al., 2015*; *Spear, 2018*). On the other hand, never satisfied with what one current has could relate to risk-taking either by 'nudging' subjects into the loss domain and all the options are framed as potential or sure losses or by elevated attention to losses (loss aversion) (*De Martino et al., 2010*; *De Martino et al., 2009*; *Kahneman and Tversky, 1979*; *Sokol-Hessner et al., 2013*; *Sokol-Hessner et al., 2009*; *Thaler and Johnson, 1990*). By taking advantage of an experimental design that included gain, loss and mixed gambles, we not only confirmed the association between greed personality trait and risk-taking in general, but also had the opportunity to demonstrate such correlation was selectively mediated by individual's loss aversion, but not risk attitude (*Figure 3—figure supplement 2*). In parallel, our neuroimaging results showed that BOLD responses in the mOFC, in addition to representing gain and loss prospects, encoded the neural loss aversion signal which in turn mediated the relationship between individual greed trait score and behavioral loss aversion coefficient (*Figure 5*). Therefore, our data provide a unique neural link to associate greed personality trait with subject's risk-taking behavior.

Although it has been suggested that trait-greed might be directly related to risk attitude (*Mussel et al., 2015*), it is worth noting that in our task greed scores did not correlate with subjects' risk attitude measurement, and this result also corresponded well with a previous study showing that greed personality was not associated with risk seeking preference in the gain-only gambles (*Seuntjens et al., 2015*). We reasoned that such discrepancy could be due to the fact that most of our subjects in the fMRI study were young adults (mean age: 21.360 years) and previous studies have suggested that adolescents and young adults tend to have higher risk tolerance (*Barkley-Levenson and Galván, 2014*; *Grubb et al., 2016*). Therefore, one possible explanation is that the risk attitudes of the young adults, whom most of our subjects were, might cluster at the ceiling where the greed personality trait related effect on risk attitude is minimal. Indeed, prospect theory model estimated parameters showed that our subjects tended to be risk neutral (mean = 1.033) instead of risk averse as commonly observed in other studies with similar experimental setups (*Suzuki et al., 2016*). Future studies of subjects with a greater range of age are required to test whether greed personality scores are related to individual risk attitudes.

In the current study, we identified that mOFC, whose activity represented a 'neural loss aversion' signal, mediated the effect of greed personality on individual behavioral loss aversion. According to prospect theory, loss aversion is another critical factor influencing subjects' risk-taking, especially in the mixed gamble environment (*Barkley-Levenson et al., 2013*; *Kahneman and Tversky, 1979*; *Köbberling and Wakker, 2005*; *Tversky and Kahneman, 1992*). Numerous studies have explored the possible neural bases for loss aversion. It has been hypothesized that loss aversion might be elicited by the interaction of activities in different brain structures. For example, some studies revealed that sensitivity to losses was driven by negative emotions such as anxiety or fear, which was associated with elevated response in the amygdala, whereas gain processing was implemented in the activities of reward pathway structures such as ventral striatum (*Camerer, 2005*; *De Martino et al., 2010*; *Sokol-Hessner et al., 2013*). However, recent studies also found that compared to healthy control subjects, unmedicated anxiety patients exhibited equivalent levels of loss aversion (*Charpentier et al., 2017*), contradicting the interaction of distributed brain structure hypothesis. The other line of research, however, suggested the possibility of encoding the subjective value of potential gains and losses in the same brain regions, such as ventral striatum and mOFC/ventromedial prefrontal cortex (vmPFC), and the asymmetric activities towards gains and losses within these regions might compose the neural loss aversion signal (*Canessa et al., 2013*; *Tom et al., 2007*). A variety of literatures have suggested that the OFC plays a pivotal role in processing rewards and punishments (*Knutson et al., 2003*; *O'Doherty et al., 2001*), making it well suited among the candidates of encoding a neural loss aversion signal. For instance, BOLD responses in the mOFC were found to correlate with subjective (both appetitive and aversive) values of different modalities of stimuli, such as smell (*Gottfried et al., 2002*), snack foods (*Plassmann et al., 2010*; *Suzuki et al., 2017*), soda drinks (*McClure et al., 2004*) and monetary gains and losses (*De Martino et al., 2009*). Consistent with previous research, we also found mOFC activity correlates with both potential gains and losses and the response asymmetry towards gains and losses in the mOFC was correlated with

behavioral loss aversion (*Figure 4D*). Furthermore, this neural loss aversion representation of mOFC mediated the effect of greed personality on behavioral loss aversion (*Figure 5*), providing additional support for the critical role of mOFC in integrating potential gains and losses and functioning as a neural network gateway to bias choice behavior (*Behrens et al., 2008*; *Kable and Glimcher, 2007*; *Kable and Glimcher, 2009*). With a more lenient threshold ($z = 2.3$, $p < 0.05$ FDR correction), we also observed activities in the ventral striatum that correlated with both gains and losses (*Figure 4— figure supplement 1*). However, unlike previous research, our ROI analysis didn't yield a significant correlation between ventral striatum activity and behavioral loss aversion, potentially due to the difference in the experimental designs (*Canessa et al., 2013*; *Tom et al., 2007*). Indeed, in our task, options were presented as the two-alternative choice selection task, in contrast to the default option accept and reject task used in earlier studies (*Tom et al., 2007*). There has been evidence suggesting the way options were presented and framed might have a peculiar effect on value representation and subsequently behavioral risk-taking (*Boureau and Dayan, 2011*; *De Martino et al., 2009*; *Wright et al., 2012*).

Finally, it is worth emphasizing that greed is a multi-faceted personality trait and may manifest as a myriad of behavioral propensities under different contexts. For example, greed might be tied to subjects' attitude towards ambiguity, the uncertainty of risk itself, or to asymmetrical updating of positive and negative outcomes (*Grubb et al., 2016*; *Hsu et al., 2005*). Future research will be required to study whether greed can be traced to unique neuroanatomical or functional representations in the brain and how it might interact with related brain networks to enable specific context dependent behavior.

## Materials and methods

### Subjects

To confirm the correlation between greed personality and impulsivity personality, we ran a pilot study of 76 university students (18–29 years old, mean = 22.513, s.d. = 2.625; 46 females) who were asked to complete the online versions of both the greed and impulsivity personality trait questionnaires.

Another 28 healthy adult subjects took part in the fMRI experiment. Three subjects were excluded from data analysis due to either falling asleep during scanning (one subject) or misunderstanding of task instructions (two subjects), leaving 25 subjects (18–26 years old, mean = 21.360, s. d. = 2.158; 11 females) for both behavioral and imaging data analyses.

Additionally, we ran a behavioral replication study by recruiting another 51 subjects to complete both the personality questionnaires and the risky decision task after the fMRI experiment. Two subjects who chose risky gambles less than 10% of all trials were excluded from data analysis, leaving 49 subjects (18–25 years old, mean = 21.200, s.d. = 2.061; 25 females) for further data analyses.

All subjects reported no history of neurological or psychiatric disorders and no current use of any psychoactive medications and had normal or corrected to normal vision. Informed written consent was obtained from each subject before the experiment. This study was approved by the Ethics Committee of the School of Psychological and Cognitive Sciences, Peking University (2017-11-01).

### Assessment of personality traits

All subjects were instructed to complete two personality questionnaires. The 7-item Dispositional Greed Scale (DGS) was used to measure individual differences in trait-greed (*Mussel et al., 2018*; *Seuntjens et al., 2015*). Subjects were asked to indicate the extent to which they agreed that those items in the questionnaire were descriptive of themselves (e.g. 'One can never have too much money'). A 5-point scale was used to rate each item with a response format ranging from 1 'strongly disagree' to 5 'strongly agree'. The total scores can range from 7 to 35, and a higher score indicates a higher level of greed personality.

Impulsivity personality was evaluated using the popular Barratt Impulsiveness Scale (BIS-11) (*Patton et al., 1995*). The trait scale (BIS-11) with 30 items was divided into three subscales: cognitive, motor, and non-planning. Each item is rated on a 5-point frequency scale ranging from 'almost never' to 'almost always' with higher scores indicating higher levels of impulsiveness trait.

## Experimental task design

During the fMRI experiment, all subjects completed a gambling task with 230 trials of four different types: gain, loss, mixed and catch trials. In the gain condition (40 trials), subjects were asked to choose between a certain reward (¥1 to ¥ 15) and a gamble with equal probabilities of receiving zero or a reward with varying magnitudes (¥ 3 to ¥ 30). We designed these trials such that the ratio of gamble reward and certain gain ranged from 1.5 to 3.4, covering a reasonable range of risk-taking according to previous literatures (*Grubb et al., 2016*; *Sokol-Hessner et al., 2013*; *Sokol-Hessner et al., 2009*; *Tom et al., 2007*). In the 40 loss trials, we simply flipped the signs of gamble and certain option amounts in the gain condition. In the 144 mixed trials, subjects were asked to choose between a guaranteed amount of zero and a gamble with equal probabilities of a gain (¥ 8 to ¥ 30, in increments of ¥ 2) and a loss (¥ 5 to ¥ 27, in increments of ¥ 2). Finally, to ensure that subjects remained actively engaged in the task, we added six catch trials (three in the gain frame and three in the loss frame) where there existed objectively optimal choices (For example: 50% chance of ¥ 10 versus ¥ 10 for sure). All the subjects made correct choices for these catch trials, suggesting continued engagement throughout the experiment. The sequence of all 230 trials, as well as gamble/certain option location (left versus right) on the screen, was randomized across subjects and trials were divided into two equal sessions. In each trial, subjects were instructed to respond within 5 s after the choice options were revealed on the screen. Once subjects registered their choices, the chosen option was highlighted with a red rectangle for 1 s; if no choice was entered during the response window, a message 'Please respond faster!' was displayed for 1 s. Subjects responded with reaction time (RT) of 2.013 ± 0.382 s (mean ±s.d.) for each trial. A randomly jittered inter-trial interval (ITI) with a mean of 6 s (range: 4–8 s) was introduced before the beginning of next trial (see experiment procedure in *Figure 1* and gamble sets in *Figure 1—figure supplement 1*).

Each subject was endowed with ¥30 at the beginning of the experiment (maximum amount subjects could have lost in any single trial), and was told that one randomly selected trial at the end of the experiment would be played for real money to enforce incentive compatibility. All the subjects also received a ¥90 subject fee upon completion of the study and they on average earned ¥124 at the end of the experiment.

## Behavioral analysis and quantitative decision model

To investigate the relationship between greed personality trait score (GPT) and subjects' gambling propensity, we adopted a multiple regression approach and controlled the potential impact of impulsivity personality trait score (IPT) by including it as a covariate of no interest.

In a more systematic approach, we modeled subjects' choice behavior following a standard three-parameter prospect theory model (*Kahneman and Tversky, 1979*; *Sokol-Hessner et al., 2013*; *Sokol-Hessner et al., 2009*; *Tversky and Kahneman, 1992*). The subjective values of gains and losses were modeled as a power function (*Equation 1*), where parameters $\alpha$ and $\lambda$ were risk attitude and loss aversion coefficients, respectively. $\alpha < 1 (>1)$ indicates risk aversion (seeking) while $\alpha = 1$ indicates risk neutrality. Similarly, $\lambda > 1 (<1)$ indicates that the individual is loss averse (gain seeking) and when $\lambda = 1$, gains and losses are valued equally (gain-loss neutral). The subjective value of a gamble is simply the expectation of the valuation of outcome variable (*Equation 2*). Subject's gambling propensity was modeled as the logistic transformation of the difference between subjective values of the gamble and the guaranteed option (*Equation 3*). Finally, parameter $\tau$ indicates subject's choice consistency (*Li and Daw, 2011*; *Sokol-Hessner et al., 2009*).

$$u(x) = \begin{cases} x^{\alpha}, & if \ x \geq 0; \\ -\lambda(-x)^{\alpha}, & if \ x < 0 \end{cases} \tag{1}$$

$$u_{gamble} = \frac{1}{2}u(outcome_1) + \frac{1}{2}u(outcome_2) \tag{2}$$

$$P_{gamble} = \frac{1}{1 + e^{-\tau\left(u_{gamble} - u_{guaranteed}\right)}} \tag{3}$$

To fit our model to subjects' choice behavior, we adopted the hierarchical Bayesian analysis (HBA) approach (*Gelman and Hill, 2006*), which has been shown to provide more reliable parameter

recoveries than conventional maximum likelihood estimation (MLE) (*Ahn et al., 2013*). Under this framework, individual-level $\alpha$, $\lambda$ and $\tau$ were samples from group level normal distributions $N(\mu, \sigma^2)$. To avoid the bias to the posterior distribution when sample size is small, normal and half-Cauchy distributions were used to specify the priors of the group-level means ($\mu$ ~Normal (0, 1)) and standard deviations ($\sigma$ ~Cauchy (0, 5)), respectively. In the model fitting, all parameters were restricted to positive values. Posterior inference for the model was performed using standard Markov-Chain Monte Carlo sampling methods implemented in rStan (v2.15.1) (*Carpenter et al., 2017*). A total of 24000 samples were drawn after a burn-in period of 12000 samples with four chains. All model parameters had R-hat values less than 1.01, indicating MCMC chains had converged to the target posterior distributions (*Ahn et al., 2017*).

We also tested a model in which subjects' risk attitudes in the gain and loss domain can be different:

$$u(x) = \begin{cases} x^{\alpha^+}, & if \; x \geq 0; \\ -\lambda(-x)^{\alpha^-}, & if \; x < 0 \end{cases} \tag{4}$$

to determine the better behavioral model and its derived regressors for further imaging analysis. Bayeisan model comparison analysis using deviance information criteria (DIC) showed that the different risk attitudes model (*Equation 4*) did not perform significantly better than the single risk attitude model: the protected exceedance probability (PXP) was 0.539 and the Bayesian omnibus risk (BOR) indicator measuring the probability that all model frequencies are indistinguishable (0 indicates the models are well distinguishable, and 1 means the models are indistinguishable) was 0.796 (*Spiegelhalter et al., 2014*; *Rigoux et al., 2014*; *Stephan et al., 2009*). We thus focused on reporting the results from the single risk attitude model.

## Imaging data acquisition and preprocessing

Whole-brain image collection was performed using a Siemens Prisma 3T scanner with a 64-channel head coil at the Center for MRI Research of Peking University. For each functional session, T2*-weighted functional images were acquired with a simultaneous multi-slice (SMS) sequence supplied by Siemens (62 slices, repetition time (TR) = 2000 ms, echo time (TE) = 30 ms, multi-band factor = 2, flip angle = 90°, field-of-view (FOV) = 224 × 224 mm, slice thickness = 2 mm, slice gap = 0.3 mm, voxel size = 2 × 2 × 2 mm$^3$, posterior to anterior phase encoding direction).

Additionally, MP-RAGE T1-weighted image for each subject was also obtained (192 slices; TR = 2530 ms; TE = 2.98 ms; flip angle = 7°; field-of-view=224×256 mm; voxel size = 0.5 × 0.5 × 1 mm$^3$). For each subject, a field map with a gradient echo (GRE) sequence was collected to correct the magnetic field distortion.

All image data preprocessing and analyses were implemented with SPM12 (Wellcome Trust Center for Neuroimaging). EPI images were first slice-timing corrected (with the middle slice as the reference slice), and then corrected for magnetic field distortion. After correcting for head movement effect, the functional images were co-registered to the T1-images which had been segmented into white matter, gray matter and cerebrospinal fluid using SPM default tissue probability maps and normalized to the standard Montreal Neurological Institute (MNI) space with final image resolution of 2 × 2 × 2 mm$^3$. Finally, normalized images were spatially smoothed using a 6 mm full-width at half-maximum (FWHM) Gaussian kernel, and temporally filtered with a high-pass filter with 1/128 Hz cutoff frequency.

## Imaging data analysis

To investigate the brain responses towards prospective gains and losses, we constructed a general linear model (GLM) with the onsets of the option revelation modulated by three parametric regressors (potential gains of the gambles, the loss magnitudes of the gambles, and the guaranteed option value). We further verified that the correlation between the magnitudes of potential gains and losses was indeed not significant ($r = 0.011$, $p = 0.867$). Additionally, six estimated head movement regressors and the onsets of choice decision were also included as covariates of no interest. The regressors in the GLM design matrix were then convolved with the canonical hemodynamic response function (HRF) with the BOLD time series in each voxel as the dependent variable. The parameter contrast estimates of each subject following the individual GLM analysis were then entered into the group

level random-effect statistical tests. All results are whole-brain corrected for multiple comparison by reporting brain activities surviving a voxel-level uncorrected threshold of $p<0.001$ and then a cluster-level false discovery rate (FDR) threshold of $p<0.05$ (*Rouam, 2013*). Using Imcalc tool in SPM12, the region of interest (ROI) in the conjunction analysis was created as the overlap of the brain activity maps positively related to increasing gains and negatively to increasing loss magnitudes (*Figure 4C*). The neural loss aversion at the ROI for individual subject was calculated by subtracting the GLM parameter estimators of potential gains from those of the losses ($-\beta_{loss} - \beta_{gain}$) (*Tom et al., 2007*).

## Mediation analysis

We carried out mediation analysis using statistical package 'PROCESS' in SPSS (http://processma-cro.org/index.html) (*Preacher and Hayes, 2008*). The neural loss aversion in the mOFC was tested as the potential mediator variable of the relationship between GPT (input variable) and behavioral loss aversion (outcome variable) (*Figure 5*). Additionally, individual IPT scores were also included in the model as covariates of no interest, which were mathematically treated as a predictor for both the mediator variable and the outcome variable. The 95% bias corrected bootstrap confidence intervals (CIs) of the indirect effect were calculated on the basis of 5000 bootstrap samples.

## Acknowledgements

JL was supported by the Chinese National Science Foundation grants: 31421003, 31371019 and National program on Key Basic Research Project grant 2015CB559200.

## Additional information

### Funding

| Funder | Grant reference number | Author |
|---|---|---|
| National Natural Science Foundation of China | 31421003 | Jian Li |
| Ministry of Science and Technology of the People's Republic of China | 2015CB559200 | Jian Li |
| National Natural Science Foundation of China | 31371019 | Jian Li |

The funders had no role in study design, data collection, and interpretation, or the decision to submit the work for publication.

### Author contributions

Weiwei Li, Formal analysis, Investigation, Writing—original draft; Haixia Wang, Xiaofei Xie, Conceptualization, Writing—review and editing; Jian Li, Supervision, Funding acquisition, Investigation, Methodology, Writing—original draft, Project administration, Writing—review and editing

### Author ORCIDs

Weiwei Li (iD) https://orcid.org/0000-0003-0990-0243
Haixia Wang (iD) http://orcid.org/0000-0001-8897-4819
Jian Li (iD) http://orcid.org/0000-0002-3941-2622

### Ethics

Human subjects: Human subjects: All participants provided written informed consent. Study procedures were reviewed and approved by the Ethics Committee at Peking University (2017-11-01).

### Decision letter and Author response

Decision letter https://doi.org/10.7554/eLife.45093.022
Author response https://doi.org/10.7554/eLife.45093.023

## Additional files

### Supplementary files

• Supplementary file 1. Tables for statistical results and model parameters.
DOI: https://doi.org/10.7554/eLife.45093.015

• Supplementary file 2. The discriminant and convergent validities of Dispositional Greed Scale (DGS) in the datasets.
DOI: https://doi.org/10.7554/eLife.45093.016

• Supplementary file 3. Tables of brain activations correlating with potential gains and losses.
DOI: https://doi.org/10.7554/eLife.45093.017

• Transparent reporting form
DOI: https://doi.org/10.7554/eLife.45093.018

### Data availability

All data generated or analysed during this study are included in the manuscript and supporting files. Source data files have been provided in https://osf.io/rpve7/

The following dataset was generated:

| Author(s) | Year | Dataset title | Dataset URL | Database and Identifier |
|---|---|---|---|---|
| Li W | 2019 | Neural mediation of greed personality trait on economic risk-taking | http://dx.doi.org/10.17605/OSF.IO/RPVE7 | Open Science Framework, 10.17605/OSF.IO/RPVE7 |

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
