## [Decision Letter]

Thank you for submitting your article "Neural mediation of greed personality trait on risk preference" for consideration by *eLife*. Your article has been reviewed by three peer reviewers, one of whom is a member of our Board of Reviewing Editors, and the evaluation has been overseen by Michael Frank as the Senior Editor. The reviewers have opted to remain anonymous.

The reviewers have discussed the reviews with one another and the Reviewing Editor has drafted this decision to help you prepare a revised submission.

Summary:

This paper examines the relationship between greed and risk taking at the behavioral and neural levels. Subjects made choices about gain, loss and mixed lotteries. Model fitting was used to estimate the risk attitude and loss aversion of each subject. Greed and impulsivity were assessed with self-report questionnaires. The authors report that greed was correlated with loss aversion, but not risk attitude, across subjects, and that the individual loss aversion fully mediated the correlation between greed and risky choice. In the brain, they identify a region within mOFC whose activity reflects a neural loss aversion signal (as defined by Tom et al., 2007, Science: (-β_loss) – β_gain) which correlates with λ (replicating Tom et al., 2007). They further show that activity in this area correlates with greed, and that the neural loss aversion signal fully mediates the relationship between greed and behavioral loss aversion.

There are a number of very positive things to say about this interesting study. All reviewers agreed that greed is an important factor in human behavior, but how it contributes to behavior and the neural mechanisms that facilitate such contribution are underexplored. The experimental design of this study is simple and straightforward, the behavioral and neural analyses are appropriate, and the results are clean and interesting. However, the reviewers identified a number of issues that need to be addressed. In particular, the reviewers agreed that the following revisions are essential for the paper to move forward.

Essential revisions:

1) Address concerns regarding robustness and replicability of the behavioral results, e.g., ideally conducting a behavioral replication sample with larger sample size.

2) Address issue of discriminant and convergent validity of the greed personality trait, along the lines of what was reported in Table 3 of Seuntjens et al., 2015.

3) The authors assume a similar curvature of the utility function (i.e. risk attitude) in the gain and loss domains. While this is an assumption of Prospect Theory, studies showed that risk attitudes in the gain and loss domain are not correlated across individuals (e.g. Abdellaoui et al., 2007). Thus, it seems important to test a model with two alphas, and compare it to the one-α model. If the two-α model describes the behavior better, it would be important do redo the analyses with the new parameters; if the one-α model prevails, then it will be enough to mention this in the text.

4) It is not clear why ln(λ) was used for correlations with mOFC activity, whereas λ was used for correlations with the GPT scores. We assume this was done following Tom et al., but they used the ratio of loss to gain slopes from the behavioral choice data (i.e., -β_loss / β_gain) and not an estimated parameter from a model. It would be important to show that GPT and ln(λ) are indeed correlated. This is particularly important in the context of the mediation effect, which is not meaningful if the direct path is not significant to begin with. Relatedly, please test whether the results replicate if the authors use the same implementation as Tom et al. (i.e., ln(-β_loss / β_gain)).

---

## [Author Response]

Essential revisions:1) Address concerns regarding robustness and replicability of the behavioral results, e.g., ideally conducting a behavioral replication sample with larger sample size.

Before the fMRI study, we ran a pilot study of 76 university students (18-29 years old, mean =22.513, s.d. =2.625; 46 females) who were asked to complete the online questionnaires of the Dispositional Greed Scale (Seuntjens, et al., 2015) and Barrett Impulsiveness Scale (Patton, Stanford and Barratt, 1995). These pilot results revealed that the greed personality scores were indeed positively correlated with their impulsivity personality scores (*r* = 0.272, *p* = 0.018), consistent with the results found in the 25 subjects in the fMRI task.

Following the reviewers’ suggestion, we further recruited another 51 subjects to complete both the personality questionnaires and the risky decision task. Two subjects who chose risky gambles less than 10% of all trials were excluded from data analysis, leaving 49 subjects (18-25 years old, mean = 21.200, s.d. = 2.061; 25 females) for further data analyses. A simple two-sample t-test analysis revealed that there was no significant difference between these 49 subjects and the 25 subjects in our original manuscript in both the greed personality score and the impulsivity personality score (Greed: *t* = 0.382, *p* = 0.704; Impulsivity: *t* = -0.518, *p* = 0.606).

Consistent with our results reported in the original manuscript, in this behavioral sample we also found that the greed personality score (mean = 22.690, s.d. = 4.736) was significantly correlated with the percentage of risky choices in the mixed trials (*r* = 0.354, *p* = 0.013) but not in gain (*r* = 0.098, *p* = 0.507) nor loss trials (*r* = 0.180, *p* = 0.222) trials, and the impulsivity personality trait (IPT) score was not correlated with the percentages of risky choices in three conditions (see Figure 2—figure supplement 2).

We also performed similar prospect theory model based analysis. Consistent with what we reported in the fMRI study, we found that the greed personality score was significantly correlated to loss aversion (*r* = -0.325, *p* = 0.024; λ: mean = 1.711, s.d. = 0.687) but not the risk attitude (*r* = -0.089, *p* = 0.547; α: mean = 1.037, s.d. = 0.182) (see Figure 3—figure supplement 3). The *r* and *p* values reported above were obtained after controlling impulsivity score (mean = 33.707, s.d. = 9.804) as a regressor of no interest, and regression analyses without controlling impulsivity score yielded similar results. Furthermore, we also found a significant negative correlation between loss aversion coefficient and the percentage of risky gamble choice across all trials (*r* = -0.897, *p* < 0.001), and individual subject’s loss aversion coefficient (λ) fully mediated the correlation between GPT score and risky choice percentage across subjects (indirect effect: a×b = 0.009, 95% CI [0.001 0.018]; direct effect: c’ = 0.002, 95% CI [-0.002 0.006]) (see Figure 3—figure supplement 4). We have summarized the statistics of the new behavioral dataset in the Supplementary file 1—Table S1.

In conclusion, the results of the pilot online questionnaires and a newly collected behavioral study both confirmed our behavioral results reported in the manuscript. We have revised the manuscript accordingly to reflect the newly reported data in both Materials and methods and Results sections.

Results:

“We first ran a pilot online study asking participants to fulfill the Dispositional Greed Scale (DGS) (Seuntjens et al., 2015) and Barratt Impulsiveness Scale (BIS-11) (Patton, Stanford and Barratt, 1995) to measure greed personality trait (GPT) and impulsivity personality trait (IPT) and tested the potential relationship between their GPT and IPT scores, as previous study suggested (Seuntjens et al., 2015).”

“Consistent with previous studies (Seuntjens et al., 2015), we found subjects’ GPT and IPT scores significantly correlated with each other in the pilot questionnaires (*r* = 0.272, *p* = 0.018) as well as in the fMRI dataset (*r* = 0.543, *p* = 0.005; Figure 2A), suggesting that IPT might be a confounding factor in identifying the specific effect of trait-greed.”

“Importantly, we replicated these results by recruiting an independent cohort of subjects and asked them to perform the same behavioral task (Figure 2—figure supplement 2).”

“(Similar results were found in the behavioral replication study, see Figure 3—figure supplement 3-4)”.

Materials and methods:

“To confirm the correlation between greed personality and impulsivity personality, we ran a pilot study of 76 university students (18-29 years old, mean = 22.513, s.d. = 2.625; 46 females) who were asked to complete the online versions of both the greed and impulsivity personality trait questionnaires.”

“Additionally, we ran a behavioral replication study by recruiting another 51 subjects to complete both the personality questionnaires and the risky decision task after the fMRI experiment. Two subjects who chose risky gambles less than 10% of all trials were excluded from data analysis, leaving 49 subjects (18-25 years old, mean = 21.200, s.d. = 2.061; 25 females) for further data analyses.”

2) Address issue of discriminant and convergent validity of the greed personality trait, along the lines of what was reported in Table 3 of Seuntjens et al.. 2015.

We have conducted the discriminant and convergent validity analyses on all 3 datasets (pilot questionnaire, fMRI, and newly collected behavioral data) and summarized the results in the following table. These results suggested that our data were on par with Seuntjens et al., 2015, and mainly replicated the validity of the greed personality trait. We have included a table in the supplementary material (Supplementary file 2) to provide more details about the discriminant and convergent validities.

3) The authors assume a similar curvature of the utility function (i.e. risk attitude) in the gain and loss domains. While this is an assumption of Prospect Theory, studies showed that risk attitudes in the gain and loss domain are not correlated across individuals (e.g. Abdellaoui et al., 2007). Thus, it seems important to test a model with two alphas, and compare it to the one-α model. If the two-α model describes the behavior better, it would be important do redo the analyses with the new parameters; if the one-α model prevails, then it will be enough to mention this in the text.

We thank reviewers for this insightful comment. To test whether a two-α model describes subjects’ behavior better, we performed the model comparison analyses between prospect theory models with one and two alphas (risk attitude). The DIC (deviance information criterion) was calculated for Bayesian model selection (Spiegelhalter et al., 2014). We found that neither model prevailed (see Author response image 1): the protected exceedance probability (PXP) for two-α model was 0.539 (Rigoux et al., 2014; Stephan et al., 2009). Furthermore, the Bayesian omnibus risk (BOR) indicator measuring the probability that all model frequencies are indistinguishable (0 indicates the models are well distinguishable, and 1 means the models are indistinguishable) was 0.796 in our task, suggesting that the one-α model performed similarly as the two-α model (Rigoux et al., 2014). Following the law of parsimony (Ockham’s razor), we reported one-α model results in the manuscript and added the information about two-α model in the Materials and methods section (subsection “Behavioral analysis and parametric decision model”).

4) It is not clear why ln(λ) was used for correlations with mOFC activity, whereas λ was used for correlations with the GPT scores. We assume this was done following Tom et al., but they used the ratio of loss to gain slopes from the behavioral choice data (i.e., -β_loss / β_gain) and not an estimated parameter from a model. It would be important to show that GPT and ln(λ) are indeed correlated. This is particularly important in the context of the mediation effect, which is not meaningful if the direct path is not significant to begin with. Relatedly, please test whether the results replicate if the authors use the same implementation as Tom et al. (i.e., ln(-β_loss / β_gain)).

We really appreciate this comment and performed further analyses as the reviewers suggested. First, the correlation between greed personality trait score (GPT) and ln(λ) was significant (*r* = -0.434, *p* = 0.034). We have now added this result into the manuscript:

“Moreover, as expected, such neural loss aversion signal also negatively correlated with individual subject’s GPT score (*r* = -0.449, *p* = 0.028; robust regression, *t* = -2.239, *p* = 0.035), echoing our behavioral results of the association between GPT scores and behavioral loss aversion (ln (λ); *r* = -0.434, *p* = 0.034)”.

Second, we also examined the relationship between loss aversion coefficients estimated from linear regression (i.e. λ_r_ = -β_loss/β_gain) and prospect theory model (λ_p_). As expected, there is a strong correlation between λ_r_ and λ_p_ (*r* = 0.807, *p* < 0.001). Moreover, the correlations between GPT and ln (λ_r)_ (*r* = -0.474, *p* = 0.019); GPT and λ_r_ (*r* = -0.464, *p* = 0.023) are both significant. However, the mediation effect with ln(λ_r_) as the output variable did not reach significance (bootstrap 5000 samples, IPT controlled: indirect effect: a×b = -0.016, 95% CI [-0.064 0.026]; direct effect: c’ = -0.064, 95% CI [-0.138 0.010]), potentially due to the marginal correlation between neural loss aversion signal in mOFC and ln (λ_r_) (*r* = 0.377, *p* = 0.063).